# MDEU-Net: Medical Image Segmentation Network Based on Multi-Head Multi-Scale Cross-Axis

**DOI:** 10.3390/s25092917

**Published:** 2025-05-05

**Authors:** Shengxian Yan, Yuyang Lei, Jing Zhang, Xiao Gao, Xiang Li, Penghui Wang, Hui Cao

**Affiliations:** Shaanxi Key Laboratory of Ultrasonics, School of Physics and Information Technology, Shanxi Normal University, Xi’an 710062, China; 2023303971@snnu.edu.cn (S.Y.); lyy_weirong@snnu.edu.cn (Y.L.); zhanggale@snnu.edu.cn (J.Z.); gao0515@snnu.edu.cn (X.G.); lixiangideal@snnu.edu.cn (X.L.); wang_penghui@snnu.edu.cn (P.W.)

**Keywords:** medical image segmentation, cross-axis attention, multi-scale features, multinomial attention, feature fusion

## Abstract

Significant advances have been made in the application of attention mechanisms to medical image segmentation, and these advances are notably driven by the development of the cross-axis attention mechanism. However, challenges remain in handling complex images, particularly in multi-scale feature extraction and fine-detail capture. To address these limitations, this paper presents a novel network architecture, multi-head multi-scale cross-axis attention MDEU-Net, that leverages a multi-head attention mechanism processing input features in parallel. The proposed architecture enables the model to focus on both local and global information while capturing features at various spatial scales. Additionally, a gated attention mechanism facilitates efficient feature fusion by selectively emphasizing key features rather than relying on simple concatenation and improves the model’s ability to capture critical details at multiple scales. Furthermore, the incorporation of residual connections further mitigates the gradient vanishing problem by enhancing the model’s capacity to capture complex structures and fine details. This approach accelerates computation and enhances processing efficiency, while experimental results demonstrate that the proposed network outperforms traditional architectures in terms of performance.

## 1. Introduction

Medical image segmentation is a fundamental task in medical image processing with broad applications in areas such as disease diagnosis [1,2], surgical planning, and lesion monitoring [3,4,5,6]. Lesions are precisely identified and located using this technique that enables more accurate diagnoses. The development of precise surgical plans to minimize risks and complications is also supported while measurements of the size and volume of organs, tissues, or lesions are provided. Additionally, it offers quantitative metrics for assessing the condition and monitoring treatment outcomes.

Currently, two-dimensional segmentation models based on convolutional neural networks (CNNs) [7,8] have become widely adopted in medical image segmentation [9,10]. One of the most prominent models is U-Net [11], which features a unique U-shaped encoder–decoder architecture and improves segmentation accuracy through the introduction of skip connections. This is particularly useful for medical image processing tasks. So, the success of U-Net has inspired the development of various network architectures, including several U-Net variants. U-Net++ [12] enhances model performance by replacing traditional connectivity with nested and dense skip connections [13,14]. Additionally, ResU-Net [15] models incorporate residual connections into U-Net to mitigate the gradient vanishing problem [16] during training and demonstrate strong performance in medical image segmentation tasks.

With the continuous development and refinement of network models. TBConvL-Net [17] combines the advantages of Transformer global modeling and Convolutional Neural Network (CNN) local feature extraction to address the instability problem in medical image segmentation. However, it still has shortcomings in extracting detailed and boundary features. Cluster Center Transformer [18] and LCAUnet [19] can effectively focus on the local edges around the region by using the attention mechanism, but they still have limitations in generalization ability and computational resource requirements. QRMFO [20] is an optimization algorithm simulating moth–flame behavior with strong global search capability and an efficient parallel mechanism. However, it may prematurely converge to local optima, failing to attain the global optimum. MCANET’s [21] attention mechanisms mitigate traditional CNNs’ long-range dependency capture limitations; however, its single-head design fails to reliably model global context, leading to performance issues.

Conventional convolutional neural networks, comprising feedforward neural networks (FNNs), convolutional neural networks (CNNs), and recurrent neural networks (RNN), alongside transformer neural networks frequently utilized in prevalent large-scale models, possess the capability to extract both local and global features within medical image segmentation applications. However, when dealing with complex and variable morphologies, it is challenging to achieve precise segmentation, especially in cases involving pathological regions or intricate organ morphology details, where its performance becomes limited. In medical image segmentation, the accuracy of segmentation is significantly affected by factors that contribute to challenges in achieving precise results, such as morphology, scale, and boundary ambiguity. As a result, developing methods to effectively capture multi-scale information in medical images [22,23] and improving model robustness and accuracy through flexible network architectures have become key research focuses in the field.

An improved method for medical image segmentation integrating cross-axis attention [21,24,25] and multi-scale feature fusion to enhance segmentation accuracy is presented in this paper. First, the introduction of the cross-axis attention mechanism enables the model to effectively capture global information from multiple directions within the image, thereby boosting the boundary recognition capabilities of complex lesion regions. Simultaneously, multi-scale feature extraction refines the feature representation of lesion regions at different scales and enhances the model’s ability to process lesions of varying sizes. Next, efficient feature fusion [26,27,28] is implemented through a gating mechanism that selectively emphasizes key features and mitigates the gradient vanishing problem via residual connections. This enhances the model’s ability to capture complex structures and details. Finally, the multi-head attention mechanism learns attention patterns across multiple subspaces in parallel to enhance the model’s robustness and segmentation accuracy for medical images with complex morphologies and ambiguous boundaries.

In summary, our main contributions can be summarized as follows:

1. Cross-axis attention mechanisms enable cross-exchange computation between x-axis and y-axis information. This achieves a more comprehensive capture of critical visual features within images.

2. Multi-scale feature extraction through hierarchical scale-based feature capture effectively accommodates structural variations in size and morphology within medical imaging data. This approach demonstrates particular advantage for segmentation of minute pathological entities requiring discriminative feature discrimination across spatial granularities.

3. Multi-head attention mechanisms perform parallel computation across independent attention heads, enabling information capture across diverse subspaces. Each head learns distinct attention patterns, and their fusion into a unified representation significantly enhances model performance.

4. Efficient feature fusion via gating mechanisms selectively attends to critical regions of input feature maps instead of mere concatenation, enabling enhanced multi-scale feature capture. Residual connections alleviate vanishing gradients by directly transmitting inputs to deeper layers.

## 2. Network Architecture

The MDEU-Net model is proposed in this paper to address the challenges of multi-scale information extraction and detail capture in complex images, and it incorporates the MDMSC Multihead Attention [29] module and the EF Efficient Feature Fusion module.

### 2.1. General Organization

The architecture of the MDEU-Net model enhances the U-Net encoder–decoder structure, as shown in Figure 1. The encoder is composed of four convolutional modules that each consist of two convolutional layers [30] followed by a ReLU activation function [31] and a max pooling [32] operation. Within the module, the feature map undergoes a 3 × 3 convolution and ReLU activation. Then, the ’same’ padding is employed to ensure the spatial dimensions of the feature map remain unchanged. Subsequently, a downsampling operation is performed, which reduces the size of the feature map by half, while the number of channels is adjusted according to the labeling shown in Figure 1. Furthermore, a multi-head, multi-scale cross-axis attention mechanism is incorporated to enhance feature extraction. The decoder also consists of four convolutional modules that each contain an upsampling layer, an efficient feature fusion module, and two convolutional layers. The key innovations of the model include the incorporation of a multi-head, multi-scale cross-axis attention mechanism at each downsampling stage, which enhances multi-scale feature modeling. The model also adopts an efficient feature fusion module that replaces the traditional U-Net skip connections and improves feature fusion and information transfer efficiency.

### 2.2. Multi-Head Multi-Scale Cross-Axis Attention Module

As shown in Figure 2a, the proposed architecture consists of eight parallel multi-scale channel (MSC) modules that extract multi-scale features from the image through downsampling and distribute the channels evenly to eight single-head attention modules. Each single-head attention module follows the MSC principle by extracting multi-scale features and then sequentially stacking the output channels of all modules. A residual connection (RC) is introduced that facilitates effective information transfer to mitigate the vanishing gradient problem and reduce information loss.

The multi-scale cross-axis attention structure illustrated in Figure 2b features a network design that consists of two parallel branches, each of which is responsible for extracting the horizontal and vertical features of the image, respectively. Each branch employs a distinct 1D convolutional kernel size to capture multi-scale contextual information along a single spatial dimension. These features are integrated and enhanced in the orthogonal spatial dimension through a cross-axis attention mechanism that facilitates the capture of both horizontal and vertical information. The core idea of this approach is to leverage the multi-layer perceptron (MLP) capabilities of convolution to construct fine-grained hierarchical feature representations and to enable comprehensive and detailed extraction of image features.

Taking the topmost branch in Figure 2b as an example, the input is $F^{*}\in R^{H\times W\times \frac{C}{8}}$ and the $F_{x}$ output [21] is(1)$$F_{x}=Conv1\times 1\left(\sum_{i=0}^{2}Conv1D_{i}^{x}\left(Norm\left(F^{*}\right)\right)\right)$$
where it denotes the one-dimensional convolution along the x-axis that we set to 1 × 7, 1 × 11, and 1 × 21 according to SegNeXt [33]. $Norm\left(\cdot \right)$ denoting layer normalization, and Conv1×1 denoting a 1 × 1 convolution. Similarly, for the bottom branch, the $F_{y}$ output can be expressed as [21](2)$$F_{y}=Conv1\times 1\left(\sum_{i=0}^{2}Conv1D_{i}^{y}\left(Norm\left(F^{*}\right)\right)\right)$$

Inspired by the self-attention mechanism of the Transformer, we have come to understand the crucial roles of the Key Matrix, the Value Matrix, and the Query Matrix in the attention mechanism. Specifically, the primary function of the Key Matrix is to extract significant feature information from the input data, essentially labeling each data point with a unique feature identifier. In contrast, the Value Matrix carries the actual content to be output or the weight information, integrating these features through weighted aggregation based on their relevance to the query. Lastly, the Query Matrix represents the current input information or the focal point of the model’s attention. We propose to compute the cross-attention between $F_{x}$ and $F_{y}$, which aims to better utilize the multi-scale convolutional features in both spatial directions. $F_{x}$ will be used as the key matrix [34] and the value matrix [35], and $F_{y}$ will be used as the query matrix [36]. The computation is as follows [21]: (3)$$F_{t}=MHCA_{y}\left(F_{y},F_{x},F_{x}\right)$$
where $MHCA_{y}\left(\cdot ,\cdot ,\cdot \right)$ denotes the cross attention along the x-axis, and $F_{t}$ represents the output obtained after performing attention calculation along the x-axis. For the bottom branch, feature extraction is performed along the y-axis in the same way as denoted below [21]: (4)$$F_{b}=MHCA_{y}\left(F_{x},F_{y},F_{y}\right)$$
$MHCA_{x}\left(\cdot ,\cdot ,\cdot \right)$ denotes cross attention along the y-axis, and $F_{b}$ represents the output obtained after performing attention calculation along the x-axis.

For the obtained $F_{t}$ and $F_{b}$, the output $F_{n}$ of the proposed multi-scale cross-axis attention can be expressed as [21](5)$$F_{n}=Conv1\times 1\left(F_{t}\right)+Conv1\times 1\left(F_{b}\right)$$

The single-head attention works as above, and the overall MDMSC output $F_{out}$ is expressed as(6)$$Fout=F+F_{m}$$
where *F* is the downsampled output. $F_{m}$ represents the feature output after the fusion of eight single-head attention mechanisms.

### 2.3. Efficient Feature Fusion Module

As illustrated in Figure 3, the EF module consists of two sub-modules: EA and EC. The EA module (gated attention mechanism) fuses low-level and high-level semantic features through skip connections and refines them using residual connections. The fused features are then passed to the EC module for further feature extraction.

The structure of the EA module is depicted in Figure 3c. Initially, group convolution is employed for intra-group feature fusion that contrasts with standard convolution and reduces computational complexity significantly. Following, the ReLU activation function is applied after convolving the input features. The low-level semantic features are then concatenated through residual connections that are transmitted via skip connections. Finally, the residual connection helps mitigate the interference of high-level semantic features on low-level features when the correlation between two input features is weak, and this helps prevent degradation in overall model performance. The computational expression is as follows [37]: (7)$$EA\left(g,x\right)=x\times \left(1+Sigmoid\left(Conv1\times 1\left(Relu\left(W_{g}+W_{x}\right)\right)\right)\right)$$(8)$$W_{g}=Relu\left(BN\left(GroupConv32\left(g\right)\right)\right)$$(9)$$W_{x}=Relu\left(BN\left(GroupConv32\left(x\right)\right)\right)$$
where sigmoid and ReLU are the activation functions, and BN is the Batch Normalization operation. GroupConv32 is the 32-component group convolution, and Conv1 × 1 is the regular convolution with a convolution kernel size of 1 × 1. In this model, g is the semantic feature obtained by up-sampling, and x is the low-level semantic feature passed by the jump connection.

The structure of the EC module is depicted in Figure 3b, and it incorporates a residual network based on the ECA from S-Unet [38]. The EC module eliminates the need for additional weight parameters through the global average pooling (GAP) layer, which consists of a GAP layer followed by a 1 × 1 convolutional layer. This significantly reduces the model’s parameter count and mitigates the risk of overfitting while improving computational efficiency. However, the GAP operation compresses the spatial dimensions of the feature map into a single global average, which results in the loss of spatial structure information and potentially causes the loss of local spatial details. The limitation is effectively addressed by introducing the residual network that enhances the model’s ability to recover spatial information.

## 3. Experiments and Discussion of Results

### 3.1. Experimental Preparation

#### 3.1.1. Experimental Data

This study utilizes the PanNuKe dataset for cell nucleus segmentation [39], which consists of 300 accurately labeled microscopic images. For model training and evaluation, the dataset is split into 260 for training and 40 for testing. The image annotations in the PanNuKe dataset were manually performed by skilled technicians who covered the precise contours and locations of the cell nuclei to ensure that the model could learn accurate segmentation information. For the retinal blood vessel segmentation task, we utilized the Retinal Blood Vessels in the Eye (RITE) dataset from the DRIVE database, which consists of 40 high-quality retinal blood vessel images. The dataset was split into 30 training and 10 testing groups for model training and evaluation. Each image is meticulously labeled with the exact contours and structural details of the blood vessels that aid the model in learning the intricate features of retinal blood vessels. Data augmentation techniques were applied during training to enhance model performance and fully utilize the dataset. These techniques expand the dataset in a way that increases sample diversity, reduces the risk of overfitting, and improves the model’s generalization ability and robustness. The images in the training set were subjected to rotation, translation, scaling, and color transformations. Additionally, data augmentation helps mitigate noise, uneven illumination, and biases in biomedical images, which in turn improves segmentation accuracy.

#### 3.1.2. Experimental Methods

All experiments were conducted using the PyTorch 2.0.0 framework on a computing platform with Python 3.8 (Ubuntu 20.04 operating system) and an Nvidia RTX 4090 GPU (24 GB graphics memory). The MDEU-Net model was trained with the Adam optimizer, a learning rate of 0.0001, momentum of 0.9, and weight decay of $1\times 10^{-8}$. The model was trained for 200 epochs on the DSB2018 dataset with a batch size of 4 and for 40 epochs on the RITE dataset with a batch size of 2. After training, After model training, the loss function curve shown in Figure 4 indicates a significant decrease in the cell nucleus segmentation task between epochs 0 and 100, stabilizing at approximately 0.04. Similarly, the ocular vascular segmentation task showed a notable reduction between epochs 0 and 30 that ultimately converged to around 0.1. The effectiveness of the MDEU-Net model was demonstrated during the training process when its performance gradually improved over time.

### 3.2. Experimental Evaluation Indicators and Results

#### 3.2.1. Evaluation Indicators

Semantic segmentation is a pixel-level task in image segmentation. Commonly used evaluation metrics include recall, mean intersection over union (mIoU), and accuracy (Acc). All of these metrics are computed from the confusion matrix. The definitions of true positive (*TP*), false positive (*FP*), false negative (*FN*), and true negative (*TN*) are provided in Table 1:

*TP*: indicates that a sample is predicted to be a positive example and the true label is a positive example.

*FN*: indicates that a sample is predicted to be a counterexample and the true label is a positive example.

*FP*: indicates that a sample is predicted to be a positive example and the true label is a negative example.

*TN*: indicates that a sample is predicted to be a counterexample and the true label is a counterexample.

The binary semantic segmentation evaluation metric can be expressed as(10)$$Acc=\frac{TN+TP}{TP+TN+FP+FN}$$(11)$$mIoU=\frac{\frac{TP}{TP+FP+FN}+\frac{TN}{TN+FN+FP}}{2}$$(12)$$Recall=\frac{TP}{TP+FN}$$

**Table 1 sensors-25-02917-t001:** Confusion matrix for classification results.

The Real Situation	Projected Results	
	Standard Practice	Counter-Example
standard practice	TP (True Positive)	FN (False Negative)
counter-example	FP (False Positive)	TN (True Negative)

#### 3.2.2. Experimental Results and Analysis

Table 2 presents the performance comparison of seven deep learning models for cell nucleus and retinal blood vessel segmentation tasks. As shown in Table 2, the MDEU-Net model significantly outperforms other medical image segmentation models in both cell nucleus and retinal blood vessel segmentation. Compared with the original U-Net model in the task of cell nucleus segmentation, MDEU-Net improves the mIoU, accuracy (Acc), and recall by 11.52%, 3.51%, and 6.19%, respectively. Improvements of 8.33%, 0.6%, and 5.59% in mIoU, accuracy, and recall are observed for retinal blood vessel segmentation, respectively. MDEU-Net achieves a 0.47% improvement in mIoU for the cell nucleus segmentation task, demonstrating its effectiveness compared with the more advanced MCANET model. For the retinal vessel segmentation task, it demonstrates improvements of 1.02% in mIoU, 0.13% in accuracy (Acc), and 2.40% in recall, respectively. These results demonstrate that MDEU-Net achieves superior segmentation performance, which is particularly evident in the recall rate. It also maintains a relatively small model parameter size and shows notable improvements in both accuracy and intersection over union (IoU).

As shown in Figure 5, the MDEU-Net model demonstrates exceptional segmentation performance in both retinal blood vessel and cell nucleus segmentation tasks while maintaining low computational complexity. This indicates that the model not only enhances segmentation accuracy but also reduces the demand for hardware resources.

As shown in Figure 6, it compares the segmentation results of the MDEU-Net model with those of other networks for cell nucleus and retinal blood vessel segmentation. MDEU-Net efficiently captures global features and accurately recognizes object boundaries and contours. Furthermore, the model excels in local feature extraction that is particularly effective in handling complex backgrounds and fine structures.

As shown in Figure 7, a comparison of the magnified prediction maps for different segmentation models in specific regions containing labeled maps that involve MDEU-Net, S-Unet, and Res-Unet is presented. The MDEU-Net model demonstrates superior contour accuracy in the nucleus segmentation magnification map, which shows a better fit with the labeled map than the other models. In retinal vascular segmentation magnification maps where MDEU-Net outperforms other models in terms of vascular continuity and detail restoration, it places particular emphasis on the precise reconstruction of capillaries and vessel termini that closely align with the labeled maps.

The multi-head attention mechanism is a critical component in modern neural networks. Unlike single-head attention, which shares query, key, and value matrices, multi-head attention assigns independent matrices to each head and enables the model to focus on different parts of the input simultaneously. The optimal number of heads is not universally fixed and varies depending on the specific task and model architecture. To evaluate the impact of the multi-head attention mechanism on segmentation performance, Table 3 presents segmentation metrics for different head configurations across the nucleus and retinal vessel datasets. It is important to note that the number of output channels in the first downsampling layer of the MDEU-NET model is 64, and that means the number of heads must be a divisor of 64 to ensure proper model operation. As shown in the table, the configurations MDEU2-NET, MDEU4-NET, MDEU8-NET, MDEU16-NET, and MDEU32-NET correspond to two, four, eight, sixteen, and thirty-two heads, respectively. Experimental results indicate that the model with eight heads achieves superior segmentation performance in both the nucleus and retinal vessel tasks compared with other configurations.

The MDEU-NET model innovatively introduces the MDMSC multi-head attention module and the EF efficient feature fusion module on top of the U-Net architecture. To evaluate the contribution of these two modules to the model’s segmentation performance, we designed an ablation experiment. Specifically, we removed each module individually and conducted segmentation experiments on the PanNuKe dataset and the RITE dataset. Each experiment employed the Adam optimizer with a learning rate of 0.0001, momentum of 0.9, and weight decay set to $1\times 10^{-8}$. The results are presented in Table 4 and Table 5, demonstrating the impact of each module on the overall segmentation performance.

Table 4 and Table 5 show the impact of the ablation experiment on the segmentation performance of the RITE dataset and the PanNuKe dataset, respectively. As can be seen from the segmentation performance results presented in the tables, the removal of either the MDMSC multi-head attention module or the EF efficient feature fusion module has an effect on the segmentation performance. To further verify the importance of these two modules, we conducted a visual analysis of the segmentation performance of the model after removing each module. These visual results reinforce the experimental evidence and also provide a more intuitive perspective for understanding the specific contribution of each component in the segmentation task. The segmentation results are shown in Figure 8.

## 4. Conclusions

This paper presents MDEU-NET, a novel medical image segmentation network based on the U-Net architecture, which incorporates several optimizations and enhancements. First, a multi-head attention mechanism is employed, enabling the model to capture information from multiple perspectives simultaneously. This significantly improves the model’s representational capacity and performance compared with single-head attention. Second, a multi-scale cross-axis attention mechanism is designed to address the variability of organs and lesion sites in medical images, enabling each attention head to focus on features across different scales and axes, thereby enhancing the segmentation process. Finally, the Efficient Feature Fusion (EF) module is introduced as a superior alternative to traditional feature fusion methods. This module integrates the attention mechanism with a gating module, enabling detailed information to be captured at multiple levels, thereby enhancing the emphasis on critical features and improving both feature extraction accuracy and segmentation performance. Experimental results demonstrate that the MDEU-NET model excels in segmentation tasks, with a significant improvement in recall rate and notable gains in accuracy and intersection ratio. 

## Figures and Tables

**Figure 1 sensors-25-02917-f001:**
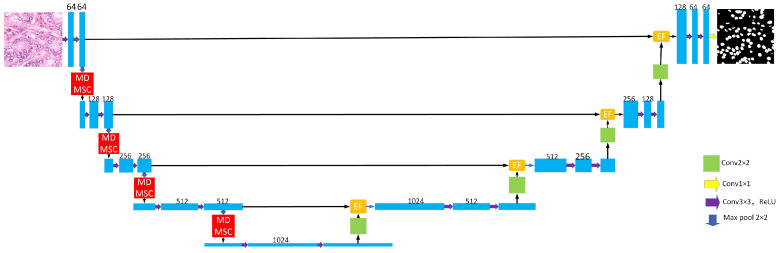
MDEU-Net network overall architecture.

**Figure 2 sensors-25-02917-f002:**
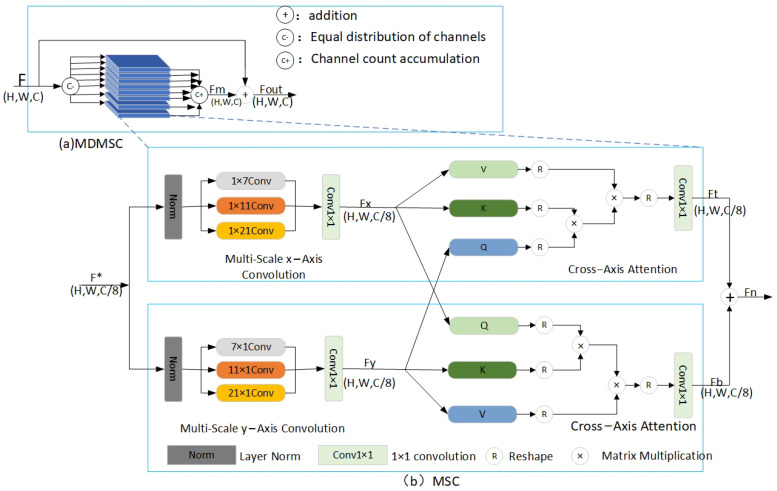
MDMSC network structure.

**Figure 3 sensors-25-02917-f003:**
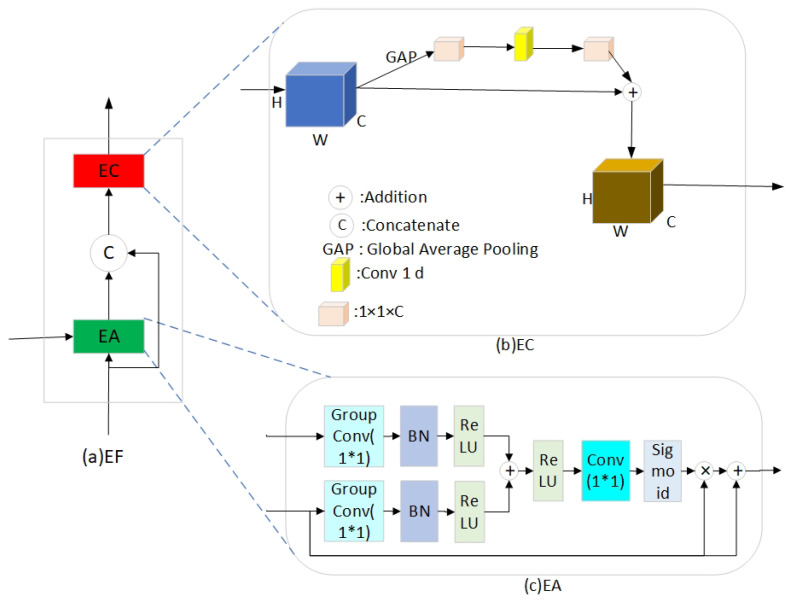
Structure of EF efficient feature fusion module.

**Figure 4 sensors-25-02917-f004:**
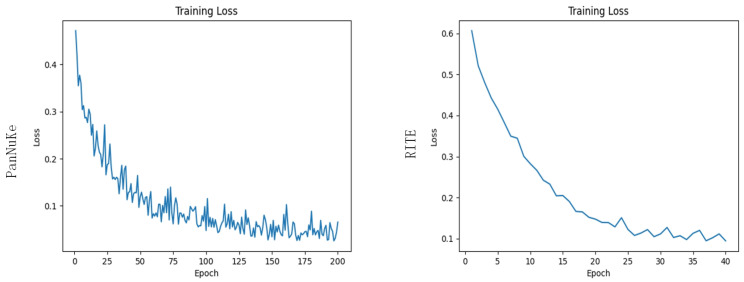
Training 200 rounds of nuclei and 40 rounds of retinal vessel loss function images.

**Figure 5 sensors-25-02917-f005:**
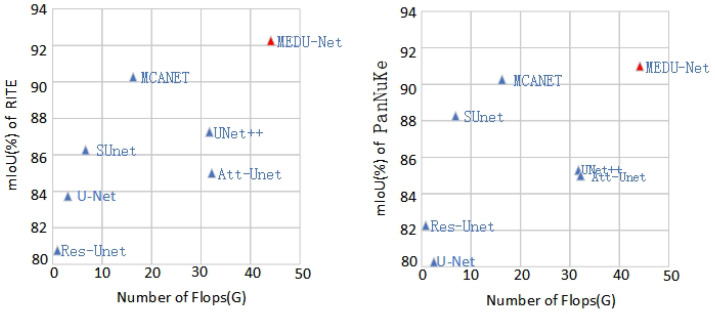
Comparison of segmentation model parameters and segmentation results.

**Figure 6 sensors-25-02917-f006:**
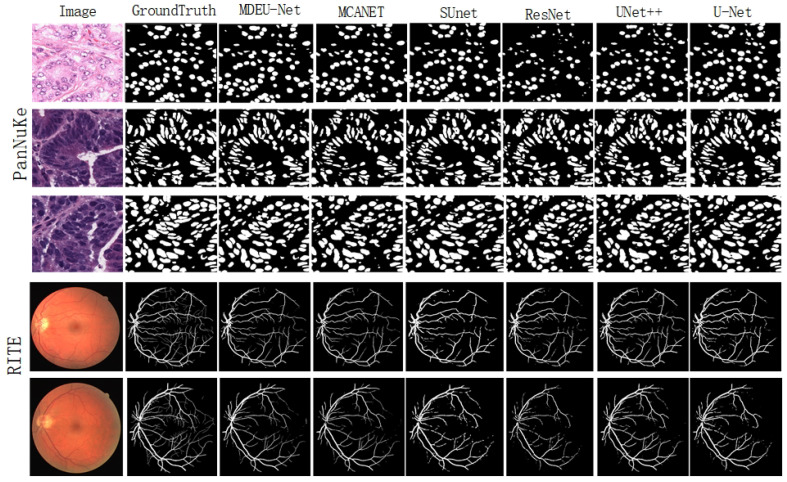
Visualization of the segmentation results of different methods for cell nuclei and retinal vessels.

**Figure 7 sensors-25-02917-f007:**
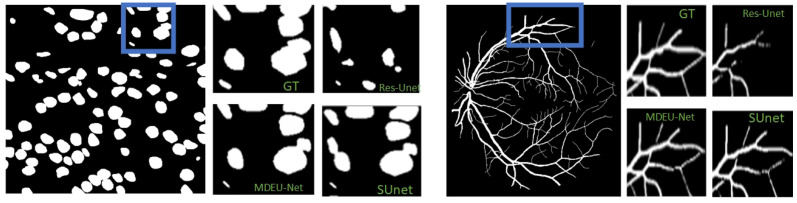
Enlarged view of different model visualizations.

**Figure 8 sensors-25-02917-f008:**
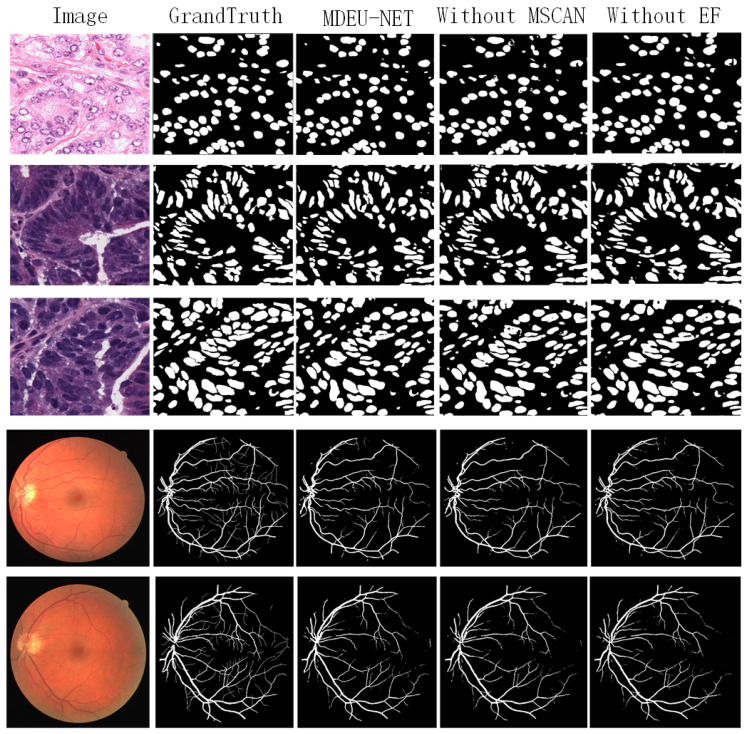
Visualization of the segmentation performance of different models after removing MDMSC and EF.

**Table 2 sensors-25-02917-t002:** Comparison of the results of cell nuclei and retinal vasculature in the MDEU-NET model and other segmentation model metrics.

	Params	Flops		RITE			PanNuKe	
Method	M	G	mIoU	Acc (%)	Recall (%)	mIoU	Acc (%)	Recall (%)
U-Net	1.56	4.08	83.77	98.17	87.81	80.01	92.67	87.23
Res-Unet	4.11	2.56	80.67	97.98	82.04	82.53	93.07	88.31
UNet++	13.41	31.13	87.40	98.67	88.37	85.42	94.19	91.06
Att-Unet	13.75	32.23	85.21	98.43	87.07	85.94	94.41	90.97
SUnet	23.01	6.31	86.74	98.53	90.24	88.00	95.30	92.82
MCANET	5.56	16.44	90.08	99.04	91.80	91.16	-	-
MDEU-Net	17.18	44.11	92.32	99.19	93.45	91.53	96.18	93.42

**Table 3 sensors-25-02917-t003:** Comparison of the results of cell nuclei and retinal vasculature in the MDEU-NET model and other segmentation model metrics.

	Params	Flops		RITE			PanNuKe	
Method	M	G	mIoU	Acc (%)	Recall (%)	mIoU	Acc (%)	Recall (%)
MDEU2-NET	17.62	44.11	91.87	99.15	93.12	90.91	95.97	93.51
MDEU4-NET	17.52	44.11	91.04	99.05	92.72	91.46	95.13	93.78
MDEU8-NET	17.18	44.11	92.32	99.19	93.45	91.53	96.18	93.42
MDEU16-NET	17.46	44.11	91.59	99.11	93.25	90.86	95.62	93.04
MDEU32-NET	17.44	44.11	90.71	99.03	91.72	89.49	95.86	92.31

**Table 4 sensors-25-02917-t004:** Impact of ablation experiment on segmentation performance of RITE dataset.

MDMSC	EF	mIoU	Acc (%)	Recall (%)
√	√	92.32	99.19	93.45
×	√	91.12	99.07	92.15
√	×	91.05	99.06	92.17

**Table 5 sensors-25-02917-t005:** Impact of ablation experiment on segmentation performance of PanNuKe dataset.

MDMSC	EF	mIoU	Acc (%)	Recall (%)
√	√	91.53	96.18	93.42
×	√	87.90	95.26	92.47
√	×	88.59	95.53	93.12

## Data Availability

The data is available upon request. Interested parties can contact the corresponding author at [2023303971@snnu.edu.cn] to obtain the dataset.

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
