# Peer review of "MDEU-Net: Medical Image Segmentation Network Based on Multi-Head Multi-Scale Cross-Axis"

_sensors, 2025, doi:10.3390/s25092917_

Round 1
Reviewer 1 Report
Comments and Suggestions for Authors
In this paper,authors proposed a novel network architecture Multi-head Multi-scale
Cross-axis Attention MDEU-Net. I have the following concerns:
1. The "gated attention mechanism" mentioned in the abstract cannot be found with specific explanations in the main text.
2. Why were eight parallel multiscale channel (MSC) modules chosen?
3. The experimental datasets have relatively small sample sizes (PanNuKe with only 300 images, RITE with only 40 images).
There is a lack of test results on more diverse medical image datasets.
4. Only three metrics (mIoU, Acc, and Recall) are used, lacking evaluation of common metrics such as Dice and F1-score.
5. Section 4 and Section 5 contain the same content.
6. What does "transaxial attention" mean in line 107?
7.The contact email in Data Availability Statement is "[email address]", and no actual contact information was provided.
Reviewer 2 Report
Comments and Suggestions for Authors
The paper "MDEU-Net: Medical Image Segmentation Network Based on Multi-Head Multi-Scale Cross-Axis" presents a novel medical image segmentation network based on the U-Net architecture, incorporating several optimizations and enhancements. To enhance clarity and relevance, the following improvements are recommended:
1. The size of the block in figure 1 is small. What is the blue block representing. Redraw, the figure for better explainability and understanding.
2. In line 38, "Traditional convolutional neural networks (CNNs) and Transformer models are widely used in medical image segmentation to capture both local and global features". The authors are suggested to add (Traditional architectures like is the suitable starting terms).
3. The dataset introduced in the article is not sufficiently justified. The authors could use other
standard datasets like ISIC, LiTS, or BRATS?
4. In line 95, the expressions are used inconsistently or not rendered, leading to readability issues.
5. The Figure 3 is not neat. The researchers are suggested to redraw it for better visibility.
6. The font-size of table content is bigger than the font size of whole article. The researchers are
suggested to keep it consistent with the whole article.
7. Keep the consistency of the words used throughout the article. For example, In line 206, As shown in Fig. 5 is written but, in the Figure 5 is written.
8. The conclusion section is redundant which seems unprofessional and unnecessary.
9. Although results are presented (e.g., Table 2), no p-values, confidence intervals, or statistical
significance tests are included to support performance claims.
10. "MEDU-Net" and "MDEU-Net" are both used in the paper. This inconsistency might confuse
readers.
11. The authors have introduced complex mechanisms like gated attention and multi-scale features could lead to overfitting, if the dataset is small and not diverse. The paper should discuss how to remove this risk by using regularization techniques.
Reviewer 3 Report
Comments and Suggestions for Authors
The manuscript proposes a novel framework but requires a revision before it can be accepted,
@ Keep citation formatting consistent. No space between words and citations.
@ Please keep a good flow of the introduction section, include importance of medical image segmentation followed by limitations of traditional CNNs. Then comes Evolution toward attention-based methods and then overview of proposed method.
@ Keep “Figure 1” and “Figure 2” inline with where they’re first mentioned in the paragraph for smoother reading.
@ In your discussion of attention mechanisms, particularly the roles of the "key matrix", "value matrix", and "query matrix", these should be clearly described in context. You can briefly mention that this approach is inspired by Transformer-style self/cross attention. For a more detailed explanation of these concepts, authors are encouraged to refer to the manuscript: DOI: 10.1109/ACCESS.2024.3506273
@ Section 2.2 (MDMSC) is very long, authors can split the section into subsections.
@ With equations, briefly explain what each variable represents.
@ Add % symbols to accuracy/recall consistently for clarity.
@ If possible, adding quantitative segmentation quality visuals (e.g., Dice plots, precision-recall curves) would further strengthen the impact.
@ During the discussion of feature fusion and attention mechanisms in the introduction, the authors should refer to recent advancements in medical image segmentation. Here are some of the most relevant recent works in the field, which could provide valuable insights and context for the proposed model. The authors may also consider searching for additional studies to further enrich this section,
- doi: 10.2174/1574893617666220920102401
- doi: 10.1109/TMM.2024.3428349
- doi: https://doi.org/10.1016/j.asoc.2024.112683
@ In the discussion of deep learning and super-resolution techniques for biomedical imaging, particularly for vascular structure segmentation, the authors should refer to recent advancements in the field to further contextualize their work. Below are some of the most relevant studies that can strengthen the discussion, with the authors encouraged to search for additional research as well:
- doi: 10.1038/s41598-024-74186-x
- doi: 10.1088/1361-6560/ad0a5a
- doi: 10.1088/1361-6560/acf98f
Round 2
Reviewer 1 Report
Comments and Suggestions for Authors
The contact email in Data Availability Statement is "[email address]", and no actual contact
information was provided.
Reviewer 2 Report
Comments and Suggestions for Authors
1) After introduction mention their atlest 4 contributions of this paper in the form of bullets.
2) The must include a seprate section for the related work and discuss all the previous SOTA models on the same problem.
3) Before the proposed architecture they must include a comaprison table that should mention strengths and weaknesses of the previous methods as well as proposed method.
4) Include the ablation study to understand the effectiveness of the proposed blocks in the selected U-Net architecture.
5) Include some good and bad cases of segmention results to understand the effectiveness of the proposed method.
Assign separate colors for the FP (red) and FN (green) pixels in the selected good and bad cases of segmentation in cell nuclei and retinal vessels.
Reviewer 3 Report
Comments and Suggestions for Authors
The majority of my comments are addressed. I just want the authors to cross-check the formatting. Especially after full stop make sure there is a space like,
'strong performance in medical image segmentation asks.With the continuous development' this should have a space before 'With'
Round 3
Reviewer 2 Report
Comments and Suggestions for Authors
Most of my comments are addressed. I recommend acceptance of this article.